# Stress-Strained State of the Thrust Bearing Disc of Hydrogenerator-Motor

**Oleksii Tretiak** [1,*], **Dmitriy Kritskiy** [2,*] , **Igor Kobzar** [3,*], **Mariia Arefieva** [1,*] , **Volodymyr Selevko** [4,*], **Dmytro Brega** [1,*], **Kateryna Maiorova** [5,*] **and Iryna Tretiak** [6,*]

1   Department of Aerohydrodynamics, National Aerospace University Kharkiv Aviation Institute, 61070 Kharkiv, Ukraine
2   Department of Information Technology Design, National Aerospace University Kharkiv Aviation Institute, 61070 Kharkiv, Ukraine
3   Special Design Office for Turbogenerators and Hydrogenerators, Join Stock Company "Ukrainian Energy Machines", 61037 Kharkiv, Ukraine
4   Section of Postgraduate and Doctoral Studies, National Aerospace University Kharkiv Aviation Institute, 61070 Kharkiv, Ukraine
5   Aircraft Manufacturing Department, National Aerospace University, Kharkiv Aviation Institute, 61070 Kharkiv, Ukraine
6   Aerospace Thermal Engineering Department, National Aerospace University, Kharkiv Aviation Institute, 61070 Kharkiv, Ukraine
*   Correspondence: alex3tretjak@ukr.net (O.T.); d.krickiy@khai.edu (D.K.); ivkobzar@ukr.net (I.K.); marii.arefieva@gmail.com (M.A.); v.selevko@khai.edu (V.S.); brega10.04@gmail.com (D.B.); k.majorova@khai.edu (K.M.); irina.ii3t@gmail.com (I.T.)

**Abstract:** In this article, the main causes of vibration in the thrust bearing of hydrogenerator motors rated 320 MW are considered. The main types of internal and surface defects that appear on the working surface of the thrust bearing disc during long-term operation are considered. A method of three-dimensional modeling of such defects is presented, and an assessment of the stress-strain state of the heel disc is proposed, taking into account the main forces acting on the working surface using the finite element method. An analysis of the possible further operation of discs with similar defects, in accordance with the technical requirements, is carried out, and we consider ways to eliminate them.

**Keywords:** hydrogenerator; thrust bearing; strength; rigidity

## 1. Introduction

Nowadays, in Ukraine, the main part of electrical power is produced at thermal, nuclear, and hydroelectric power stations, where turbo- and hydrogenerators work, respectively. Vertical-type hydrogenerators stand out among them. At the same time, an important element of the design of the generator, which perceives vertical loads from the weight of the entire unit, is the thrust bearing.

High-power generators have quite significant external geometric dimensions and consist of structural elements of various scales, which complicate (and often makes impossible) strength analysis of the generator design as a whole. At the same time, the elements of the generator structures work under complex load conditions caused by the joint action of inertial forces from the rotation of the rotor, gravitational forces, component loads arising from the seating of parts with tension, as well as temperature loads that arise, first of all, as a result of the release of heat in the active circuit and are determined by the operating parameters of their forced ventilation system. In complex generator design, this leads to the need to consider a whole complex of problems related to the determination of the thermally stressed state of structures, complicated by previous stresses, the influence of temperature fields that depend on the operating parameters of ventilation systems, and many other factors.

Figure 1 shows the design of the thrust bearing on adjustable screw supports [1]. The thrust bearing is located in an oil bath with water-oil coolers. The moving part of the support structure consists of a hub, which is mounted on the upper end of the shaft, and a ring into which the segments are inserted. The lower part of the hub is equipped with a disc with a polished surface, namely a mirror of the thrust bearing. All the vertical load is transmitted through the mirror of the bracket to the segments. The segments of the thrust bearing are arranged in one or two rows on the rest seat of the thrust bearing that accepts the load. The steel segment consists of the body of the segment and a cushion covered with a layer of anti-friction material, namely babbit or fluoroplastic. The segment rests on a plate (segment support), which is supported by spherical heads of bolts, which ensure their self-installation during operation. To prevent rotation (turning), the segments are held by special radial rests.

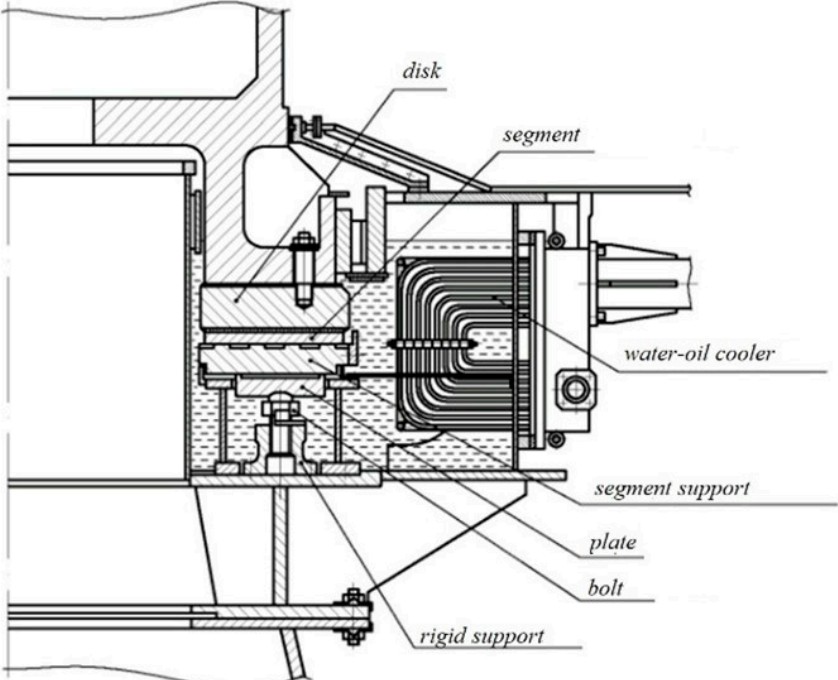

**Figure 1.** The design of the thrust bearing rest is on adjustable screw supports.

During the operation of hydro-aggregates, it is necessary to monitor the vibration of the bearings, and in the event of a vibration that is significantly greater than usual, it is necessary to make sure that the magnitude of the vibration does not exceed the permissible level before connecting the generator to the network. Otherwise, start-up is stopped, and the generator is stopped until the causes of the increased vibrations are identified. After reaching the rotation speed, which is close to the rated one, the generator is switched on for parallel operation in the network. Activation can be carried out by exact synchronization or self synchronization. Accurate synchronization assumes that the generator is connected to the network at the moment when the voltage of the generator and the network are the same in magnitude, and the voltage vectors of the generator and the network coincide in phase. This occurs when the frequency of the generator coincides with the frequency of the network. However, due to the fact that the frequency of the network constantly changes within small limits due to fluctuations in the load in the system, there is practically no long-term coincidence of the frequencies of the generator and the network. Therefore, switching on is carried out with a large or small deviation from the moment of complete coincidence of the vectors.

In work [2], the author's team arranged a detailed review of the structures of geometric forms of thrust bearings and evaluated the effectiveness of the structures. The work highlights the main problems of modeling this complex node and offers methods that

take into account the hydrodynamics and incompressibility of substances, considering the influence of temperatures. However, this work is appropriate for the simulation of new bearings; the possibility of reusing discs after long-term operation is not considered.

In work [3], a detailed review of the design of thrust bearing pads with self-adjusting pads for inclination, which is similar to the thrust bearing studied in our work, is submitted. A detailed technique of three-dimensional modeling of hydraulic and thermal processes on the surface of the segments is presented, which makes it possible to estimate the temperature effect and forces on the surface of the disc, as a result of which, defects may appear and develop on the mirror surface.

It is necessary to pay attention to the fact that the damage that occurs on the mirror surface of the thrust bearing disc as a result of operation causes an increase in the vibration of the structure.

The work [4] provides a detailed review of damaged bearing elements as a result of overheating and established cause-and-effect relationships between vibration, overheating, and the subsequent failure of the unit. The work [5] presents in detail the types of failures of hydro-aggregate units and analyzes the economic losses caused by the failure of at least one unit. The presented work is an overview and really reveals many factors affecting the appearance of additional vibrations; however, its use is limited by the lack of a three-dimensional formulation of the task under consideration.

## 2. Research Task

The purpose of this work is the analysis of the structure, the determination of typical defects, and the development of a methodology for calculating heel pads with various types of defects on the mirror surface.

The work considers a synchronous three-phase reversible hydrogenerator motor of the vertical type. Its main elements are the stator and the rotor. The stator consists of a welded body, active steel with a winding, a set of foundation plates and studs. The rotor includes a shaft, a core, a stacked-up rim, a pole with a winding, and a current supply with slip rings. In addition, the hydrogenerator motor includes: a spider consisting of a central part with an oil bath of a guide bearing, bearing segments, and oil coolers; feet of spacer jacks, overlapping of the upper cross-piece of the hydrogenerator motor; an oil bath with a thrust bearing, consisting of a bath with a seal, a thrust bearing housing with supports and segments, a thrust bearing disc, and oil coolers; main and neutral terminals of the stator winding; stand with traverse, excitation bus-bars; covering of the pump-turbine shaft consisting of beams and segments; water and oil pipelines with fittings and control and measuring devices; the ventilation system, consisting of air coolers with branch-pipes, upper and lower air distribution shields, as well as the braking system, consisting of brake jacks with stands, brake piping, a high-pressure pump, and a braking cubicle.

At the same time, it should be noted that the mass of the structural elements is, namely, the stator is 540 t, the rotor is 780 t, and the spider is 93.4 t.

## 3. Types of Surface Defects

As mentioned above, the main cause of surface and internal defects are the vibration of the structure and the influence of temperature. The review work [6] shows the most common types of surface defects that occur on the working surface of thrust bearings. This work provides comprehensive knowledge about the cause-and-effect relationships of the occurrence of defects of a certain type. The work [7] not only provides information on the causes of defects, but also contains recommendations for their elimination.

However, in this work, only the main types of defects that are most often met in the practice of disc operation are considered. The main surface defects considered are torn surfaces, hairs, hot cracks, chains (belts), bubbles, and others (see Table 1).

**Table 1.** Basic types of defects studied in this work.

| No. | 1 | 2 | 3 | 4 |
|---|---|---|---|---|
| Description | surface tear | hot cracks | gas bubbles | hair |
| Drawing | | | | |

## 4. Feature of Long-Term Operation of the Hydro-Aggregates

During the operation of the hydrogenerator, there are vibrations of the rotor caused by the load of the thrust bearing disc, and its damage leads to a significant increase in vibration of the structure in general. Figure 2 presents the results of the study of the beating of the friction surface of the thrust bearing disc with decomposition into harmonic components by the Fourier method [8].

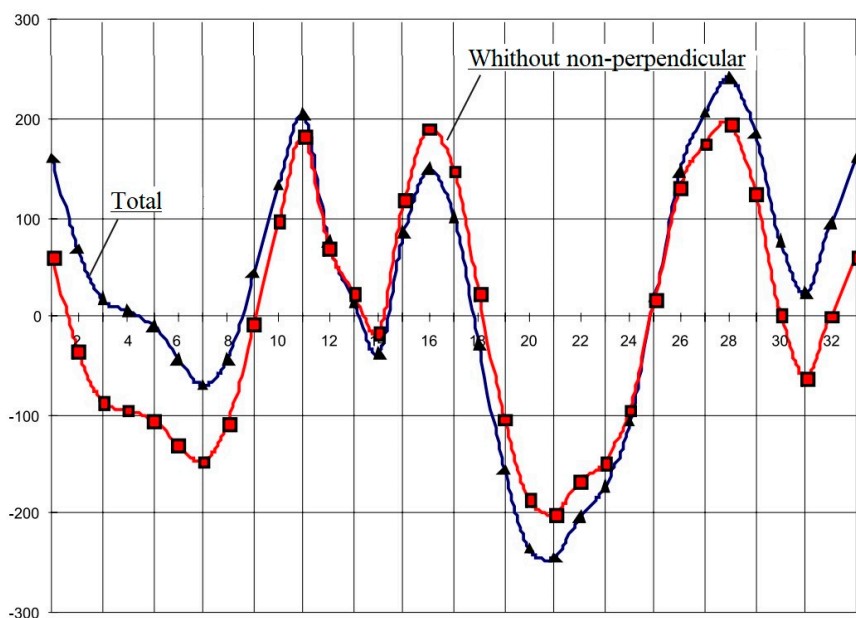

**Figure 2.** Harmonic components of vibrations of the thrust bearing disc due to beating.

However, the permissible vibration (for oscillations with a frequency of up to 100 Hz in the established symmetrical mode), which is regulated by the vibration of the frontal parts of the stator winding in the tangential and radial directions, should not exceed 100 μm [9].

## 5. The Main Causes of Vibration

The presence or absence of vibration of the hydro-aggregate determines the possibility of long-term reliable operation of the unit and is one of the main quality indicators of its design, manufacturing technology, and installation work. The increased vibration of the hydro-aggregate can lead to an emergency state, a decrease in efficiency (coefficient of useful action), and additional power losses. Therefore, when the vibration of the unit exceeds the permissible values, the causes of the increased vibration shall be established and eliminated.

The causes of increased vibration of the hydro-aggregate, depending on the source of the disturbing force, can be divided into three types: mechanical, hydraulic, and electrical.

The mechanical causes include: unbalance of the generator rotor and turbine impeller; incorrect condition and position of the shaft axis of the hydro-aggregate; problems in bearing units; weak fastening of the supporting parts of the unit or their insufficient rigidity; involvement of rotating parts of the unit with stationary parts [10].

Hydraulic causes are: hydraulic imbalance of the impeller; incorrect height position of the impeller of the radial-axis turbine in relation to the guide apparatus; incorrectly established combinatorial dependence in rotary vane turbines; turbine operation in cavitation modes [11].

The electrical causes of vibration of the unit are usually unevenness of the attraction of the rotor to the stator (electromagnetic imbalance), which is mainly caused by unevenness of the air gap of the generator, exciter, and subexciter; the oval shape of the generator rotor; and short-circuits of the windings of the rotor poles [12,13].

In the above-mentioned works, the main structural shortcomings of heel pads are presented, while the problem of vibrations in the framework of cause-and-effect relationships is not covered. However, in [14], this problem is considered taking into account modern modeling methods.

## 6. Conditions for Calculating the Stress-Strained State of the Thrust Bearing Disc

In order to calculate the stress-strained state, the mechanical loads of the mirror surface of the thrust bearing disc with long-term service defects were determined [15]. Taking into account the above, all the denominations and designations of the defect are characterized according to DSTU 2658-94 [16].

The technological requirements for the mirror surface of the thrust bearing disc during its manufacture [17] are indicated below:

1.  The roughness of the mirror surface of the disc should be no more than 0.32 μm (class 9) and not less than 0.16 μm (class 10). In some places, which make up no more than 10% of the mirror surface of the disc, the permissible purity is 0.63 μm (8-th grade).
2.  Measurements of the roughness of the mirror surface of the thrust bearing disc shall be carried out during overhauls of the unit, as well as when signs of deterioration of the mirror surface cleanliness appear (increase in the temperature of all segments at a constant oil temperature in the thrust bearing bath, etc.).
3.  The roughness of the mirror surface of the thrust bearing disc under operating conditions can be checked by removing casts on plastic material (for example, oil-gutta-percha mass) and then examining them under a microscope or using a profilometer.
4.  If the roughness of the mirror surface of the disc is worse than specified in point 1, and if there are a large number of scratches, the mirror surface of the disc shall be processed (superfinishing and subsequent polishing) and brought to 0.32 μm.
5.  The mirror surface of the thrust bearing discs of umbrella-type hydrogenerators can be processed under operating conditions with the help of a special self-propelled machine installed in the thrust bearing, from which the segments have been removed. The machine can be made for every size of thrust bearing.
6.  The mirror surface of the disc of the hanger of the suspended hydrogenerators is processed in the conditions of hydroelectric power plants with the help of simpler devices after removing the bush with the disc from the shaft. If possible, the disc is sent to the factory for machining.

Summarizing the above, during long-term operation of the thrust bearing disc, there may be cases when, after mechanical processing during major repairs, as well as friction during work, internal defects shall be revealed.

The basic document on the production of forgings is presented in [18], where it is stated that these defects may appear as a result of mechanical effects on the part.

On the machined surfaces of forgings, individual defects are allowed without removal, if their depth, determined by a control cut or scraping, does not exceed 75% of the actual

one-sided machining allowance for forgings produced by forging and 50% for forgings produced by dyeing.

## 7. Mechanical Calculation of the Disc

Conditions for mechanical calculation:

- Vertical load on to the thrust bearing disc is 1600 tf;
- Speed is 53.6 rpm.

Since the surface of the disc with clearly expressed defects interacts with the fluoroplastic coating of the heel segments, the friction force Ftr = 16000 N, the friction coefficient $f$ = 0.0001 (actual 0.05) appears accordingly. In the calculation, it is assumed that the defects are located at a uniform distance from each other and do not have mutual influence (see Figure 3).

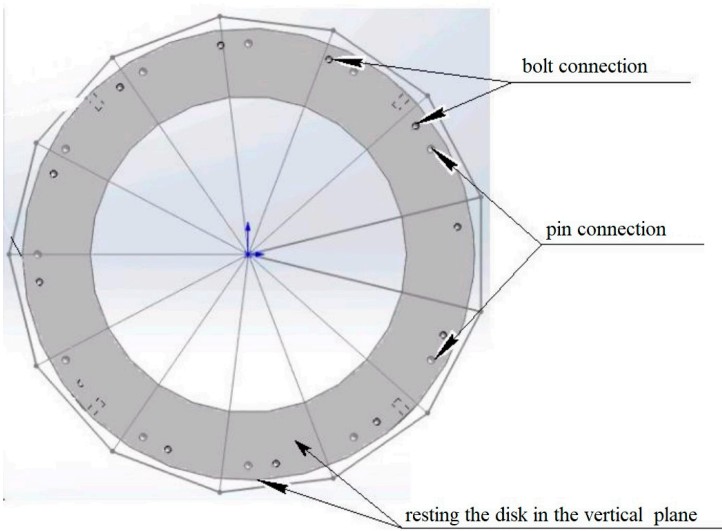

**Figure 3.** Calculation model of the thrust bearing disc.

One of the defining moments in solving finite elements method problems is the choice of a finite element. Two types of tetrahedrons (Figure 4) with different approximations of movements inside the element are used as basic finite elements. The first tetrahedron has units at the vertices (Figure 4a) and is based on a linear approximation of movements inside the element, and the second is an oblique tetrahedron that has units at the vertices of the element and in the middle of its ribs; it is based on a quadratic approximation of movements inside the element (Figure 4b).

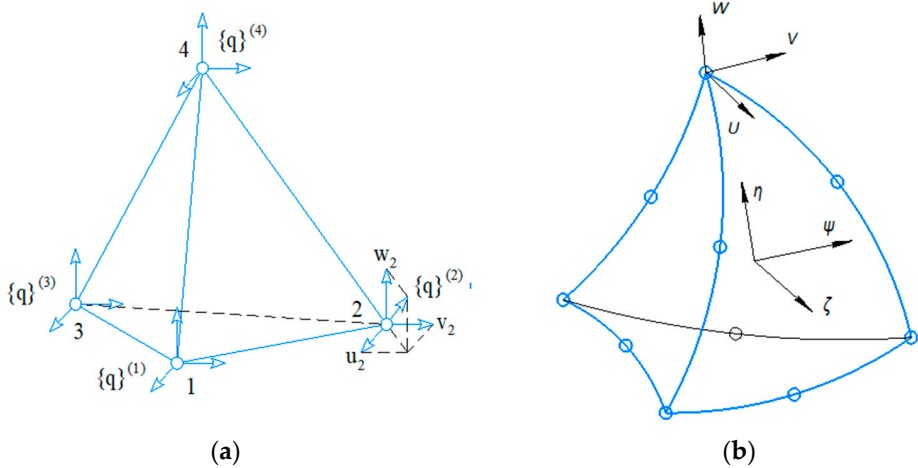

(a)　　　　　　　　　　　　　　　　　　　(b)

**Figure 4.** A finite element in the form of a tetrahedron.

An oblique tetrahedron allows for a more accurate description of the geometry and deformation process of the research object; however, it has 10 internal units and contains 30 unknown values, which is almost three times the number of unknowns for an ordinary tetrahedron with four units and, accordingly, with 12 sought values.

Therefore, a classical tetrahedron shall be used for a qualitative study of stress-strained state parameters, and an oblique one shall be used for more accurate, final calculations. In SolidsWork, which is used to solve all problems, these are TETRA4 and TETRA10 finite elements, respectively [19].

The calculation grid is built for each individual calculation of the disc section. In places where there are defects, a grid control element is used.

At the same time, there shall be at least three grid elements for the minimum geometric element. Convergence of the task is carried out by reducing the grid in such a way that the results do not differ by more than 0.5%. The accepted thickness of the oil film is 0.05 mm. Due to the fact that the depth of defects exceeds this value in the defect zones, additional forces will occur on the end surfaces.

Symmetry conditions are set on the face end surfaces of the disc segments to simplify modeling. For all defects, forces are set on the contact surface of the disc in the area of defect development. The disc-surface contact of the segment is specified as forces. At the same time, the calculation coefficient of friction in a pair (disc-fluoroplastic through oil film) according to the methodology of JSC "Ukrainian Energy Machines" is 0.05, and defects make up a very small part of the total mirror surface; then the accepted coefficient of friction for end forces is taken equal to 0.0001.

The material of the disc is forging, strength group KP245 with a yield strength of 245 MPa. In accordance with DSTU 9182:2022, defects, flakes, and cracks on the surface of the forging are permissible no more than the depth of machining. The thermal loads on the disc are set from the condition of contact heat exchange from the end surfaces of the disc and the bushing, provided by the pretension of the M48 bolts. The temperature of the disc in the summer is 30 °C, and the maximum drop is no more than 150 °C (boundary conditions of the first kind for thermal calculation). For the calculation, the disc material is set as isotropic. The change in the modulus of elasticity with temperature changes is not taken into account.

### 7.1. Defect No. 1 Surface Tear

Calculation results of stress for defect No. 1 are shown in Figures 5–9.

According to the obtained results, the average stress along the mirror surface of the disc comprises 50 MPa. In the defect location zone, the maximum stress is 625 MPa and the average stress is 520 MPa. These values exceed the strength limit (470 MPa, according to DSTU 9182:2022), yield strength (245 MPa, according to DSTU 9182:2022), and permissible stresses for rotating parts (233 MPa, according to IEC 60034-33:2022) [18,20].

### 7.2. Defect No. 2 Hot Cracks

Calculation results of stress for defect No. 2 are shown in Figures 10–13.

According to the obtained results, the average stress along the mirror surface of the disc comprises 50 MPa. In the defect location zone, the maximum stress is 901 MPa and the average stress is 750 MPa. These values exceed the strength limit (470 MPa, according to DSTU 9182:2022), yield strength (245 MPa, according to DSTU 9182:2022), and permissible stresses for rotating parts (233 MPa, according to IEC 60034-33:2022) [18,20].

According to the obtained results, the average stress along the mirror surface of the disc comprises 50 MPa. In the defect location zone, the maximum stress is 728 MPa and the average stress is 500 MPa. These values exceed the strength limit (470 MPa, according to DSTU 9182:2022), yield strength (245 MPa, according to DSTU 9182:2022), and permissible stresses for rotating parts (233 MPa, according to IEC 60034-33:2022) [18,20].

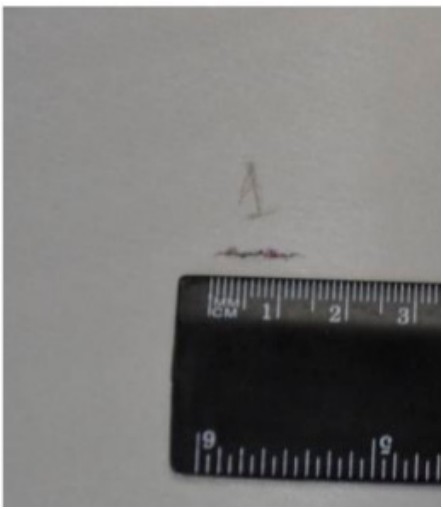

**Figure 5.** Actual defect and calculation grid for a defect on a disc segment.

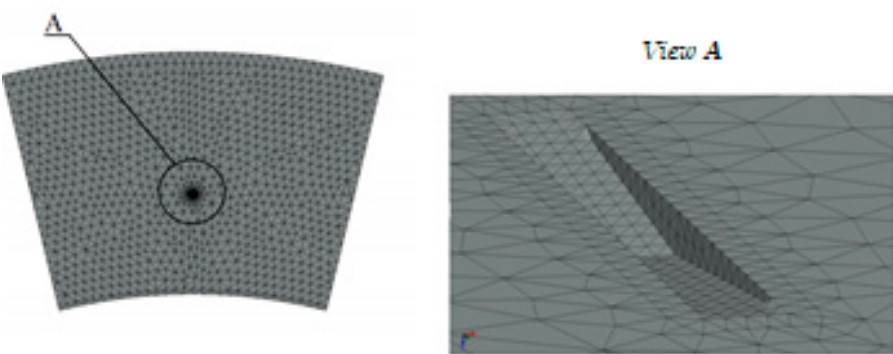

**Figure 6.** Conditions of the disc fastening.

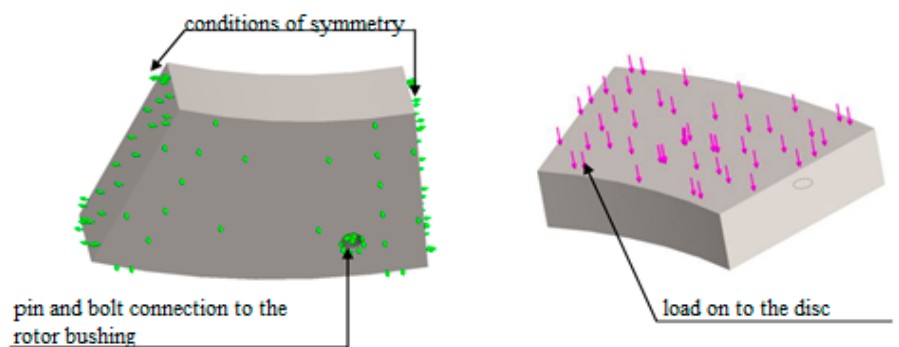

**Figure 7.** Acting load on the mirror surface of the disc.

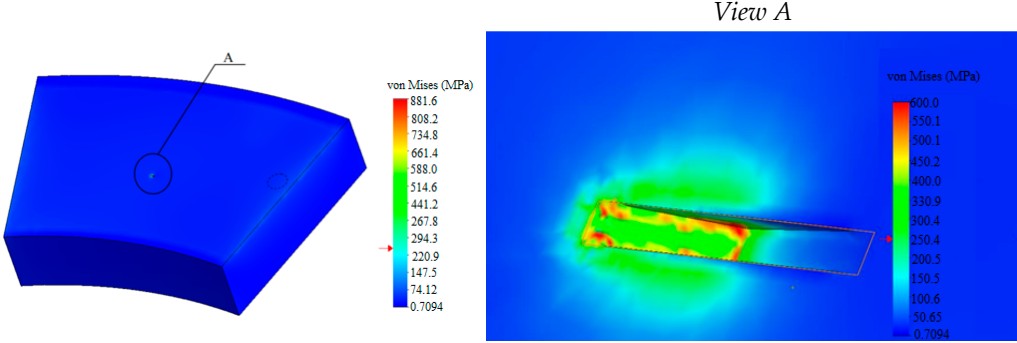

**Figure 8.** The stress field in the part of the thrust bearing disc and defect No. 1.

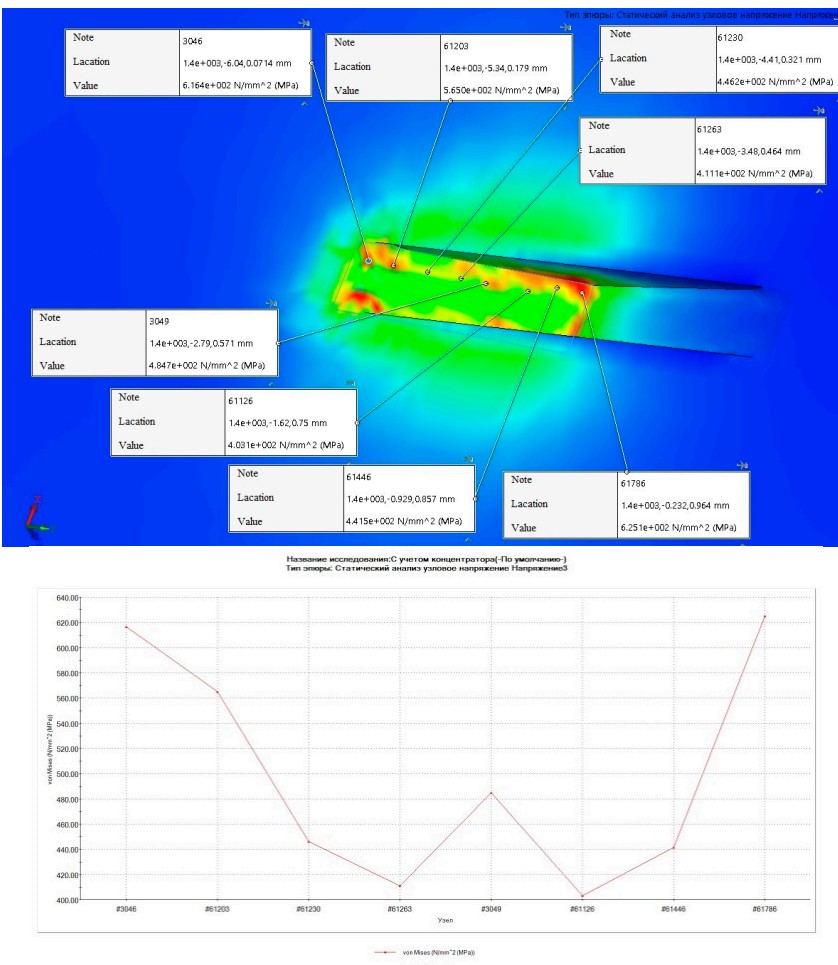

**Figure 9.** Change in stresses along the length of defect No. 1.

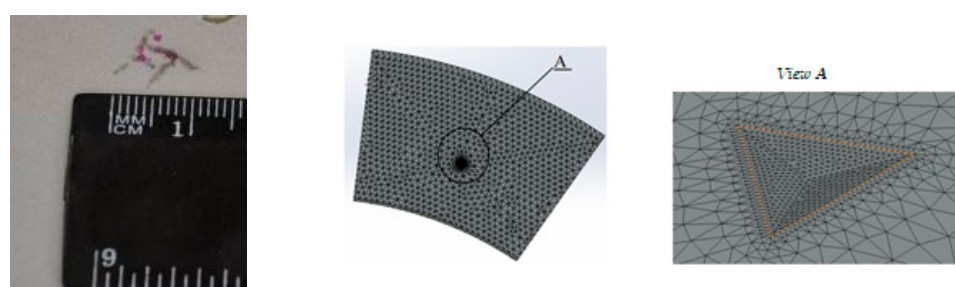

**Figure 10.** Actual defect and calculation grid for a defect on a disc segment.

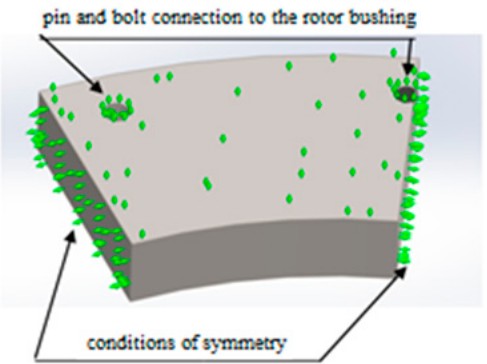

**Figure 11.** Conditions of the disc fastening.

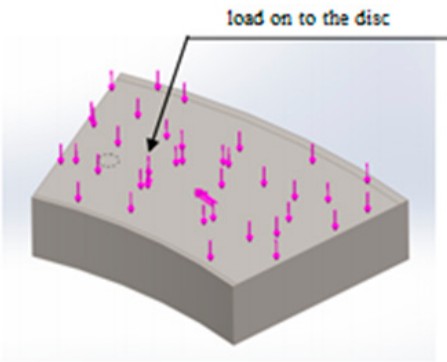

**Figure 12.** Acting load on the mirror surface of the disc.

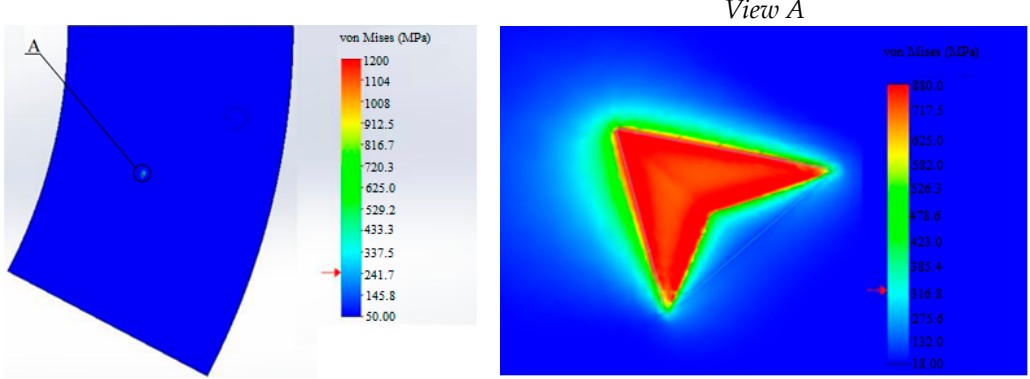

**Figure 13.** The stress field in the part of the thrust bearing disc and defect No. 2.

*7.3. Defect No. 3 Gas Bubbles*

Calculation results of stress for defect No. 3 are shown in Figures 14–18.

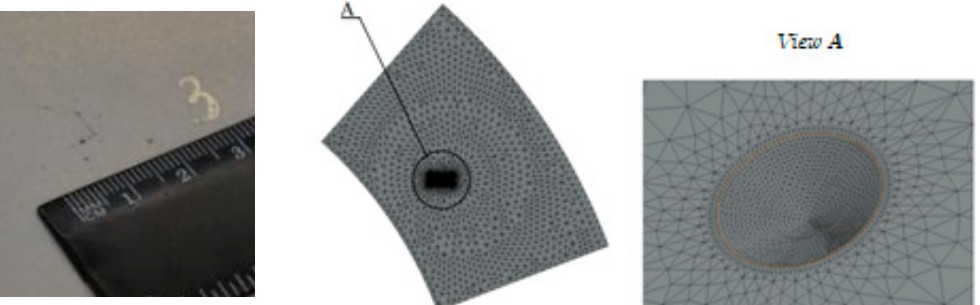

**Figure 14.** Actual defect and calculation grid for a defect on a disc segment.

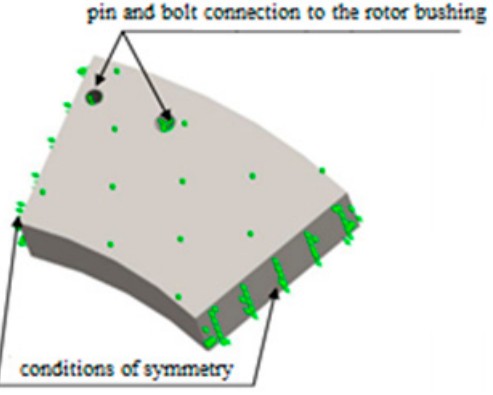

**Figure 15.** Conditions of the disc fastening.

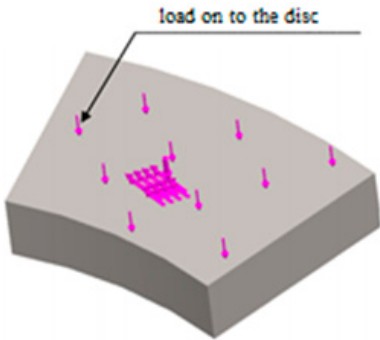

**Figure 16.** Acting load on the mirror surface of the disc.

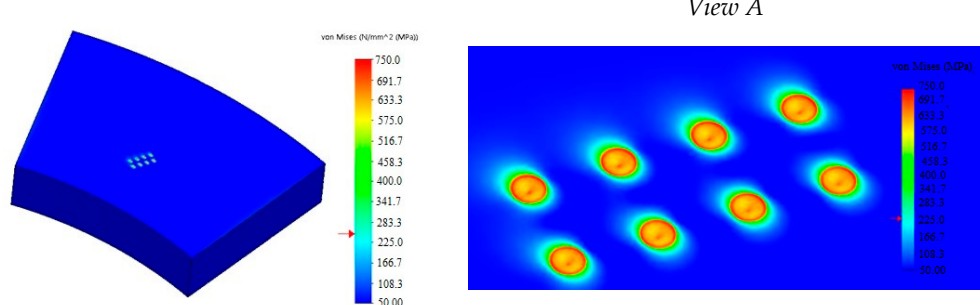

**Figure 17.** The stress field in the part of the thrust bearing disc and defect No. 3.

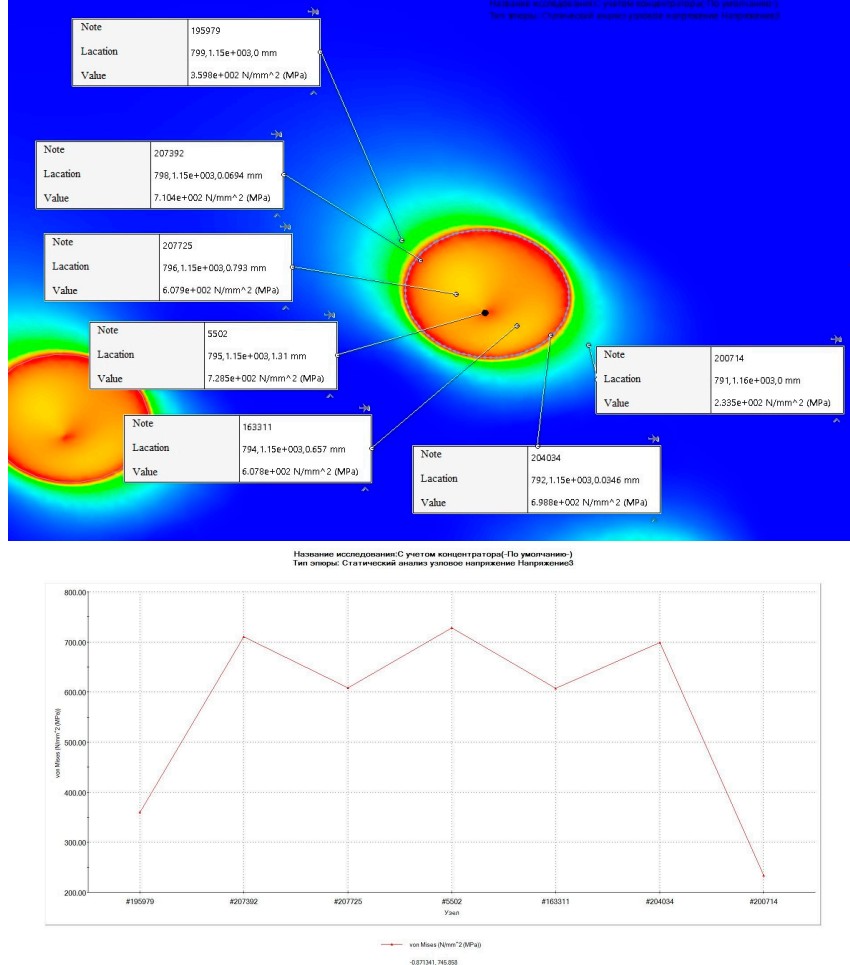

**Figure 18.** Change in stresses along the length of defect No. 3.

### 7.4. Defect No. 4 Hair

Calculation results of stress for defect No. 4 are shown in Figures 19–22.

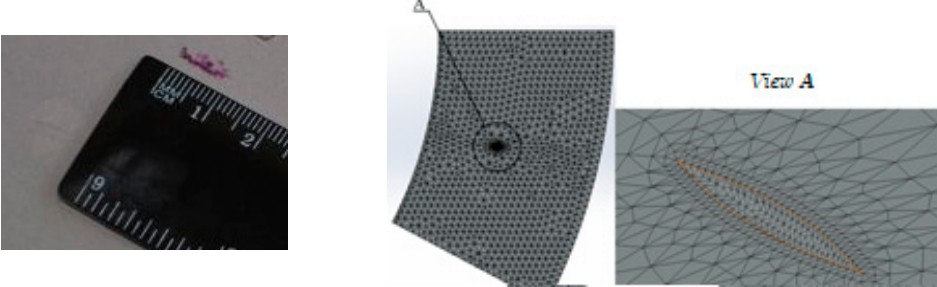

**Figure 19.** Actual defect and calculation grid for a defect on a disc segment.

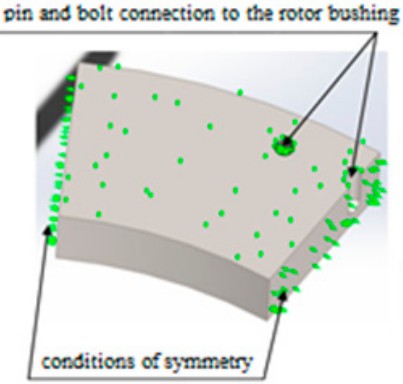

**Figure 20.** Conditions of the disc fastening.

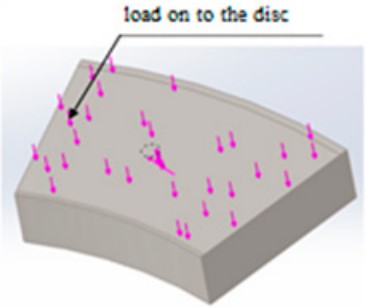

**Figure 21.** Acting load on the mirror surface of the disc.

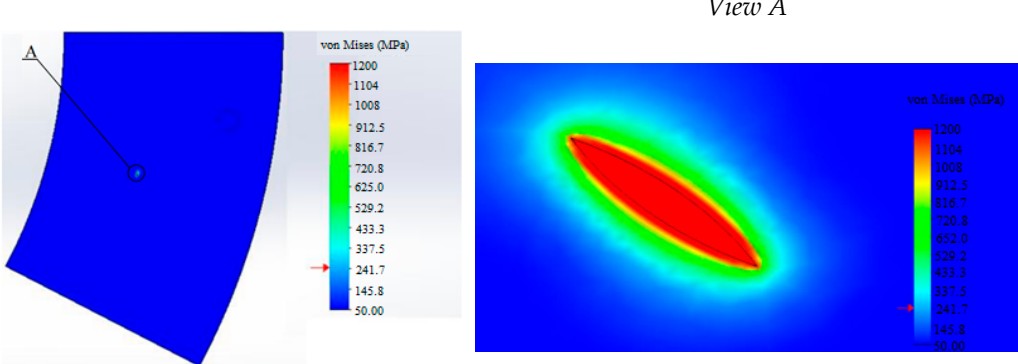

**Figure 22.** The stress field in the part of the thrust bearing disc and defect No. 4.

According to the obtained results, the average stress along the mirror surface of the disc comprises 50 MPa. In the defect location zone, the maximum stress is 1873 MPa and the

average stress is 1150 MPa. These values exceed the strength limit (470 MPa, according to DSTU 9182:2022), yield strength (245 MPa, according to DSTU 9182:2022), and permissible stresses for rotating parts (233 MPa, according to IEC 60034-33:2022) [18,20].

## 8. Discussion

As a result of the carried-out calculations, for all defects, the stress values in the defect zone exceed the permissible yield strength $\sigma t = 245$ MPa for the material from which the disc is made (forging KP 245). Namely, they exceed the permissible $0.95\sigma t$ from the yield point for rotating parts of hydrogenerators (IEC 60034-33:2022) [20], the permissible $2/3\sigma t$ from the yield point for parts of the rotor of the hydrogenerator in the rated operational mode, and $0.9\sigma t$ from the yield point at runaway speed. Due to the fact that the calculated stresses exceeded the strength limit, the issue of fatigue was not considered.

The obtained stress values in the defect zones indicate the possibility of their further development, while the operating mode of 700 cycles per year (no more than four times per day) cannot be definitely ensured.

Removal of such defects will require a significant reduction in the height of the thrust bearing disc, which in the process of operation shall lead to an increase in the effect of temperature deformations and the appearance of gaps between the rotor bush and the thrust bearing disc. In these gaps, conditions shall arise for the initiation and development of cavitation of the contact surfaces, namely, for the appearance of microcracks and microexplosions. Therefore, it is not recommended to use a thrust bearing disc with such defects.

**Author Contributions:** Writing—review and editing O.T., D.K., I.K., M.A., V.S., D.B., K.M. and I.T.; All authors have read and agreed to the published version of the manuscript.

**Funding:** This research received no external funding.

**Institutional Review Board Statement:** Not applicable.

**Data Availability Statement:** Data not available due to confidentiality.

**Conflicts of Interest:** The authors declare no conflict of interest.

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
