# Peer review of "Stress-Strained State of the Thrust Bearing Disc of Hydrogenerator-Motor"

_computation, doi:10.3390/computation11030060_

Round 1
Reviewer 1 Report
This paper studied the stress-strained state of the thrust bearing disc of a hydrogenator-motor. Some internal and surface defects are considered. Types of internal and surface defects are considered. It is an interesting work. However, there are still some issues to improve this manuscript.
1. In Figure 1, the important components are not clearly shown. The components’ names are suggested to be marked in the figure. Moreover, some enlarge view for the studied thrust bearing can be added.
2. In Introduction, the authors only listed the contributions from the previous works. For a journal paper, a literature review is suggested to be added. Moreover, the differences between the previous works and authors’ works should be clearly discussed to show the authors’ new contributions in the studied problems.
3. In Section 5, the main causes of vibrations are discussed. Some more indepth physical descriptions can be discussed. The abrupt changes of the stress and deformations at the defect edges should cause the large vibrations as given in ‘Dynamic modelling of the defect extension and appearance in a cylindrical roller bearing’. The above materials and references can be mentioned here to describe the relative issues.
4. In Section 7, the meshing element can be shown in Figures. Moreover, although some conditions are shown in the figures, more details for the FE model should be added in text, such as the material model, element type, constraints, et al.
5. All the assumptions used in the FE model should be clearly given too.
6. Many list references are too old. Some more recent references can be added to show the new contributions in the studied field.
Reviewer 2 Report
Dear Authors,
The subject considered in this manuscript is a very difficult one, therefore some clarifications, additions and corrections are necessary.
1) Illegible Fig. 1, additionally without references with names of bearing elements - It is absolutely necessary to include an additional drawing with the analysed bearing (readable), because it is not really known what type of bearing was used (tilting pad or other?) - Please see example drawings of thrust bearings by other authors [a, b, c] - (list of missing literature below the opinion - generally available on the Internet).
2) Note to Table 1 - too few cases of defect types are mentioned (even the basic ones) - No reference to literature - Recommended literature - [a, b, c].
3) There are no literature items (for example d, e, f] relating to FEM calculations of thrust bearing segments (pads), which gives the impression that only the Authors of this manuscript dealt with this issue.
4) The Authors have chosen an interesting and very difficult case of a segment surface defect for FEM analysis, therefore a detailed description of this surface discontinuity and the calculation method are required (the FEM system used - professional or own, type of finite element, how the authors dealt with the contact issue, etc. ).
The above requirements are also important because the manuscript will be published in COMPUTATION journals.
4) Hardly legible captions on the charts.
Sincerely, Reviewer
Missing literature items:
[a] William Strecker. Troubleshooting Tilting Pad Thrust Bearings. Machinery Lubrication (3/2004)
[b] A GENERAL GUIDE TO THE PRINCIPLES, OPERATION AND TROUBLESHOOTING OF HYDRODYNAMIC BEARINGS – Kingsbury, Inc. https://www.kingsbury.com/pdf/universe_brochure.pdf
[c] Seckin Gokaltun, Scan DeCamillo. COMPUTATIONAL ANALYSIS OF THE EQUALIZATION BEHAVIOR OF THRUST BEARINGS WITH REGULAR AND MODIFIED LEVELING PLATES. Proceedings of ASME Turbo Expo 2019: Turbomachinery Technical Conference and Exposition GT2019 June 17-21, 2019, Phoenix, Arizona, USA
[d] Dimitrios G Fouflias, Anastassios G Charitopoulos, Christos I Papadopoulos, Lambros Kaiktsis and Michel Fillon. Performance comparison between textured, pocket, and tapered-land sector-pad thrust bearings using computational fluid dynamics thermohydrodynamic analysis. Proc IMechE Part J: J Engineering Tribology 12-2015=4, 1-22.
[e] XL_ThrustBearing®: A Computational Physics Analysis Tool for Tilting Pad Thrust Bearings (both regular and self-equalizing types). https://rotorlab.tamu.edu/TRIBGROUP/TPTB%20Koosha.html
[f] D. Markin, D.M.C. McCarthy ∗, S.B. Glavatskih. A FEM approach to simulation of tilting-pad thrust bearing assemblies. Tribology International 36 (2003) 807–814.
Reviewer 3 Report
The paper Stress-strained state of the thrust bearing disc of Hydrogenerator-Motor brings an interesting theme, but it should be improved significantly.
The abstract is very short and does not present the area of the research sufficiently.
The introduction should focus much more on an overview of references that dealt with similar issues and only then define the own problem addressed in the article.
The Fig. 1 should be of better quality and it would be convenient to include the labeling of the individual parts of the hydrogenerator, which would significantly clarify the problem being solved in the chapter 2 - research task.
Fig. 3 would be appropriate to add a detail from which it will be more obvious how the bolt and pin connections are made.
I miss a lot of information in the chapter 7, the type of the calculation software, element types used, the size of the elements and the solver type etc. This chapter should be definitely be broaden to show the authors scientific contribution to the problem solving.
The discussion is also very short - it should also explain more in detail the results of the analyses and their verification to the reality, if possible. The Conclusion chapter would be also very suitable to supplement.
Round 2
Reviewer 1 Report
N/A
Reviewer 3 Report
Dear authors, thank you for incorporation of my comments. Now, the article will be more clear for the readers.